# Chlorophyll α Fluorescence Parameters as an Indicator to Identify Drought Susceptibility in Common Bush Bean

**Alefsi David Sánchez-Reinoso**, **Gustavo Adolfo Ligarreto-Moreno and Hermann Restrepo-Díaz ***

Departamento de Agronomía, Facultad de Ciencias Agrarias, Universidad Nacional de Colombia, Carrera 30 No. 45-03, Bogotá 11321, Colombia

* Correspondence: hrestrepod@unal.edu.co; Tel.: +57-316-5000

**Abstract:** The common bean is susceptible to drought conditions and the evaluation of plant responses to low water availability can be difficult. The quantification of chlorophyll fluorescence as a sensitive trait to environmental stresses is an important alternative in the characterization of drought-susceptible genotypes. The objective of this study was to evaluate mainly the use of chlorophyll α fluorescence (maximum efficiency of PSII ($F_v/F_m$), photochemical quenching (qP), non-photochemical quenching (NPQ)) and rapid light-response curves (RLCs) (initial slope of the curve ($\alpha$), minimum saturation irradiance ($I_k$) and maximum relative electron transport rate ($ETR_{max}$)) parameters as tools for the identification of susceptible or tolerant bush bean cultivars to water deficit stress conditions in two different phenological stages. Using a randomized block design in a factorial arrangement, five bush bean cultivars (Cerinza, Bachue, NUA35, Bacata and Bianca) were evaluated under water deficit conditions by the suspension of irrigation for 15 days from 40 to 55 Days after Emergence (DAE) (vegetative stage) or 50 to 65 DAE (reproductive stage). The results showed that $F_v/F_m$ and NPQ recorded the highest variation due to water deficit conditions, especially in the vegetative stage. The greatest reductions in $F_v/F_m$ (0.67) and NPQ (0.71) were evidenced in cultivar NUA35 compared to its control plants (0.78 and 1.07, respectively). The parameters obtained from RLCs showed that cultivar Bacata registered the lowest $\alpha$ (0.17) and $I_k$ (838.19 $\mu mol \cdot m^{-2} \cdot s^{-1}$) values compared to its control plants ($\alpha$ 0.23; $I_k$ 769.99 $\mu mol \cdot m^{-2} \cdot s^{-1}$). Differences were only obtained in $ETR_{max}$ in the reproductive stage (50–65 DAE) in which cultivar NUA35 reached values of 158.5 in stressed plants compared to control plants (251.22). In conclusion, the parameters derived from RLCs such as $\alpha$ and $I_k$ can be used as tools to identify drought susceptibility in the vegetative stage, whereas $ETR_{max}$ can be used in the reproductive stage. In addition, PSII photochemistry ($F_v/F_m$ and NPQ) can also help to understand the agronomic responses of common bush bean cultivars to drought conditions.

**Keywords:** light saturation point; *Phaseolus vulgaris*; quenching; rapid light-response curves

## 1. Introduction

The common bean (*Phaseolus vulgaris* L.) is the most cultivated and consumed grain legume in the world and plays an important role in the human diet due to its high protein and mineral content [1,2]. A worldwide production of 55,627 Mt in 38,038,865 ha was recorded for green and dry beans in 2017 [3]. Studies have projected a reduction in climate conditions due to heat and drought stress in a large part of bean crops in South America [4].

Plants are naturally exposed to different abiotic and biotic stress conditions [5]. Drought is the most decisive abiotic stress for crop yield and productivity, demanding a continuous knowledge of plant responses to this condition [6,7]. Plants under water deficit stress show a wide range of responses that include morphological, physiological, biochemical and molecular changes [8]. In addition, the decrease

in chlorophyll content under drought stress has been considered a typical symptom of oxidative stress and may be the result of pigment photooxidation and photosynthetic pigment degradation [5].

The monitoring, identification and quantification of plant responses to drought are highly demanded in breeding programs for the selection of tolerant and high-yielding genotypes [9]. In general, research on drought tolerance assessment is focused on plant survival [6]. Chlorophyll fluorescence is a highly informative technique of plant traits to cope with adverse environmental conditions [10]. In this sense, chlorophyll fluorescence uses information about the photochemical activity of plants, allowing the early detection of environmental stress [11]. This can be done because the chlorophyll molecule is fluorescent, which makes it possible to detect changes in electron transfer at the level of chloroplast membranes through photon dissipation [12]. One of the great advantages of this technique is that it is sensitive to disorders of the photosynthetic cell membrane without destroying the plant tissues of the species under study [13]. Rosenqvist and van Kooten [14] state that any variation in the values between 0.79 to 0.84 indicates that the PSII reaction centers are being affected. These authors also mention that photochemical and non-photochemical quenching (qP and NPQ) are parameters that help to understand acclimation mechanisms in plants under different environmental conditions. Finally, Flowers et al. [15] evaluated chlorophyll $\alpha$ fluorescence parameters ($F_v/F_m$, qP) in different snap bean genotypes under ozone stress, concluding that these parameters can help to determine snap bean sensitivity to high $O_3$.

Specialized equipment such as modulated fluorometers allows constructing Rapid Light-Response Curves (RLCs) which can be used to measure quantum yield in terms of irradiance [16]. From RLCs, parameters such as the initial slope of the curve ($\alpha$), minimum saturation irradiance ($I_k$) and maximum relative electron transport rate ($ETR_{max}$) can be determined, indicating the photosynthetic efficiency of plants under stress conditions [17]. RLCs is a technique that has been used for characterization of plants under drought stress in in vitro or field conditions [18,19].

Knowledge of the mechanisms of genotype acclimation to adverse environmental conditions is an effective strategy to reduce vulnerability to climate change [4]. Techniques to facilitate the identification of drought-tolerant genotypes have become important in modern agriculture [7]. Therefore, the objective of this research was to evaluate the use of chlorophyll fluorescence parameters as a tool for the identification of susceptible or tolerant bush bean (*Phaseolus vulgaris* L.) cultivars to water deficit stress conditions in two different phenological stages (vegetative and reproductive) as a trait of interest in legume improvement programs.

## 2. Materials and Methods

### 2.1. Plant Material and Growth Conditions

Seeds of five bush bean cultivars were used in the present study: (i) 'ICA-Cerinza' and 'ICA-Bachue' (cultivars with at least 20 years of sowing history in traditional Colombian agriculture); (ii) 'NUA35' (cultivar with eight years of commercialization since its release); and (iii) 'Bianca' and 'Bacata' (recently released cultivars). Seeds were sown in the greenhouse of the Faculty of Agricultural Sciences at Universidad Nacional de Colombia located in the city of Bogotá at a height of 2556 m.a.s.l., (4°35′56″ N and 74°04′51″ W) from September 2015 to January 2016 in 2.1 m$^2$ plots (3 rows of 1 linear meter long). The plant spacing was 16 cm × 70 cm between plants and between rows, respectively (20 seeds per plot = 85,000 plants ha$^{-1}$). The physical and chemical characteristics of the soil in the greenhouse were: (i) sandy loam soil (0.26 sand, 0.42 silt, and 0.32 clay); (ii) chemical characteristics: Total N 0.36%, Ca: 10.6, K: 0.98, Mg: 1.75, and Na: 0.24 meq 100 g$^{-1}$, Cu: 1.67, Fe, 310, Mn: 3.21, Zn: 15.5, B: 0.48, and P: > 116 mg kg$^{-1}$; (iii) pH 5.4; and (iv) effective cation exchange capacity (ECEC) 13.8 meq 100 g$^{-1}$. Growth conditions in the greenhouse during the experiment were: a natural photoperiod of 12 h with a Photosynthetic Active Radiation (PAR) of 1000 µmol m$^{-2}$ s$^{-1}$, average temperature of 27 °C and 60 to 80% relative humidity. Two edaphic fertilizers were used per plant: (i) 4 g/plant (350 kg ha$^{-1}$) of a 15–15–15 compound fertilizer (TRIAN 15® Yara, Colombia) as a nitrogen, phosphorus and potassium

source; (ii) and 2 g plant$^{-1}$ (170 kg ha$^{-1}$) of microelements (granulated Agrimins, Colinagro®, Colombia) respectively, divided into two applications (half of the aforementioned dose at 15 and 40 days after emergence (DAE)). Finally, the composition of the Agrimins fertilizer was: Total N 8% (Urea N 7% and Ammonium N 1%), Assimilable P (P$_2$O$_5$) 5%, Ca (CaO) 18%, Mg (MgO) 6%, Total S 1.6%, Cu 0.75%, B 1%, Mo 0.005% and Zn 0.25%.

## 2.2. Water Deficit Treatment

Three different groups of treatments were established in each cultivar: (i) plants irrigated throughout the crop cycle (control); (ii) plants with water deficit in the phenological stage 13–14 (vegetative) according to the BBCH (Bayer, BASF, Ciba-Geigy and Hoechst) scale [20] (formation of three to four fully expanded trifoliate leaves), which was reached at approximately 40 DAE; (iii) plants with water deficit during the reproductive stage according to the BBCH scale 63–64 (30 to 40% open flowers), which was reached at approximately 50 DAE. Water deficit was established by the total suspension of irrigation for 15 days. This period of water deficit stress was based on previously developed studies with variables such as photosynthesis and electrolyte leakage showed greater affectations [16]. Before and after the water deficit period, plants were irrigated in the morning (700 h to 900 h) with an equal amount of water. In the initial growth stages (up to 13–14 BBCH), plants were irrigated with 6 mm per week. During stages 15 to 55 BBCH, 12 mm were supplied per week. Finally, 18 mm were supplied per week from stage 56 to 89 according to the BBCH scale. All cultivars were irrigated at the same time with a watering can. The soil moisture content was constantly monitored by a humidity probe (Kelway HB−2 Soil Acidity and Moisture Tester, Kel Instruments Co., Inc. NJ, USA) at a depth of 20 cm during the stress period.

## 2.3. Photosynthetic Pigments and Chlorophyll Fluorescence Determination

The measurements of all photosynthetic pigments and chlorophyll fluorescence variables were taken at the end of each stress period (at 55 DAE for the vegetative stage and 65 DAE for the reproductive stage). In general, variables were estimated in the second fully expanded trifoliate leaf from the upper part of the canopy. Regarding the quantification of photosynthetic pigments, 500 mg of plant material were collected and homogenized with liquid nitrogen. Subsequently, samples were stored at −80 °C until their respective analysis. Finally, the experiment lasted approximately 115 days after the emergence of all seedlings.

Approximately, 30 mg of the stored samples were homogenized in 6 mL of acetone (80%) and centrifuged (Model 420101, Becton Dickinson Primary Care Diagnostics, MD, USA) at 5000 rpm for 10 min to remove particles [21]. The chlorophyll content was estimated at 663 and 646 nm, whereas carotenoids were determined at 470 nm.

A spectrophotometer (Spectronic BioMate 3 UV-vis Thermo, Madi-son, WI, USA) was used for both determinations. The content of leaf photosynthetic pigments was determined by the equations for acetone according to Wellburn [22].

The maximum efficiency of PSII ($F_v/F_m$), photochemical quenching (qP), non-photochemical quenching (NPQ) and rapid light-response curves (RLCs) were determined at 55 and 65 DAE using a modulated fluorescence chlorophyll meter (MINI-PAM, Walz, Effeltrich, Forchheim, Germany). Leaves were dark-adapted with leaf clips for 15 min. After dark adaptation, the fluorescence variables $F_0$, $F_m$, $F_v/F_m$, qP, and NPQ were determined. The minimal fluorescence ($F_0$) was recorded with modulated low intensity light (<0.1 $\mu$mol·m$^{-2}$·s$^{-1}$) without affecting the variable fluorescence. The maximal fluorescence ($F_m$) was estimated by a 0.8 s long saturating light pulse (2600 $\mu$mol·m$^{-2}$·s$^{-1}$) with 20,000 Hz frequency. The variable fluorescence was calculated by the difference between $F_0$ and $F_m$. $F_v/F_m$ ratio was obtained from the $F_v$ and $F_m$ and represent potential maximal PSII quantum yield. The photochemical and non-photochemical quenching were calculated as $qP = (F_m' - F)/(F_m' - F_0)$, and $NPQ = (F_m - F_m')/F_m'$ [23].

Rapid light-response curves were constructed by plotting the electron transport rate (ETR), qP and NPQ versus the increasing actinic irradiance (from 1 to 1795 $\mu mol \cdot m^{-2} \cdot s^{-1}$) with intervals of 10 s between irradiance levels [24]. The parameters $\alpha$, $ETR_{max}$ and $I_k$ were estimated using the model described by Xu et al. [17].

### 2.4. Experimental Design and Data Analysis

A randomized block design with a factorial arrangement was performed, with the phenological stage in which drought stress was initiated as the first factor and the five evaluated genotypes as the second factor, for a total of 10 treatments with four repetitions and 60 plots of 2.1 m$^2$ (3 rows of 1 linear meter long). Subsequently, when there were significant differences in the ANOVA, the comparative Tukey's test was used at $P \leq 0.05$. Data were analyzed using the Statistix v 9.0 software (analytical software, Tallahassee, FL, USA). The results of the analysis of variance are summarized in Table 1.

**Table 1.** Summary of the analysis of variance of the effects of water stress on the physiological behavior of five bush bean genotypes.

| Parameter | Abbreviation | Variation Source | | | | | |
|---|---|---|---|---|---|---|---|
| | | Vegetative Stage (55 DAE) | | | Reproductive Stage (65 DAE) | | |
| | | Stage | Cultivar | Stage × Cultivar | Stage | Cultivar | Stage × Cultivar |
| Total Chlorophyll | *Chl total* | *** | *** | *** | NS | *** | * |
| Carotenoids | *Cx + c* | *** | *** | * | ** | *** | *** |
| Maximum efficiency of PSII | $F_v/F_m$ | *** | ** | ** | * | NS | NS |
| Photochemical quenching | qP | NS | NS | NS | NS | * | NS |
| Non-photochemical quenching | NPQ | NS | * | * | *** | *** | NS |
| Initial slope | $\alpha$ | NS | ** | ** | NS | * | NS |
| Maximum electron transport rate | ETRmax | NS | NS | NS | *** | *** | *** |
| Light saturation point | $I_k$ | NS | NS | ** | NS | * | NS |

*, **, and *** significantly different at 0.05, 0.01 and 0.001 probability levels, respectively. NS, not significant with $a = 0.05$; DAE, Days after Emergence.

## 3. Results

### 3.1. Chlorophyll and Carotenoids Contents

Figure 1 shows the results obtained in the quantification of the total chlorophyll and carotenoid content. In general, it can be observed that significant differences were obtained in the interaction Stage × Cultivar on the concentration of these evaluated photosynthetic pigments for the two sampling points. In the first sampling point (55 DAE), it was observed that the majority of control plants showed a leaf chlorophyll content around 2166.26 $\mu g$ $mg^{-1}$ Fresh Weight (FW). It is important to point out that there was a greater reduction of the total chlorophyll content in cultivar Bacata plants that were subjected to water deficit stress conditions during the vegetative stage (1517.65 $\mu g$ $mg^{-1}$ FW). In the reproductive stage, cultivar Bacata continued to show the lowest chlorophyll values compared to cultivar Bachue, which presented the highest values (1538.90 $\mu g$ $mg^{-1}$ FW and ~1709.29 $\mu g$ $mg^{-1}$ FW, respectively) (Figure 1A). Regarding the carotenoid content, a reduction in this group of photosynthetic pigments was observed especially in cultivars Bianca, Bacata and NUA35 compared to their controls (~306.91 $\mu g$ $mg^{-1}$ FW vs. 362.33 $\mu g$ $mg^{-1}$ FW) due to water stress in the vegetative stage. At 65 DAE, cultivar Bachue plants under water deficit in the reproductive stage also showed a higher total carotenoid concentration compared to their control (320.50 $\mu g$ $mg^{-1}$ FW vs. 230.39 $\mu g$ $mg^{-1}$ FW, respectively) (Figure 1B).

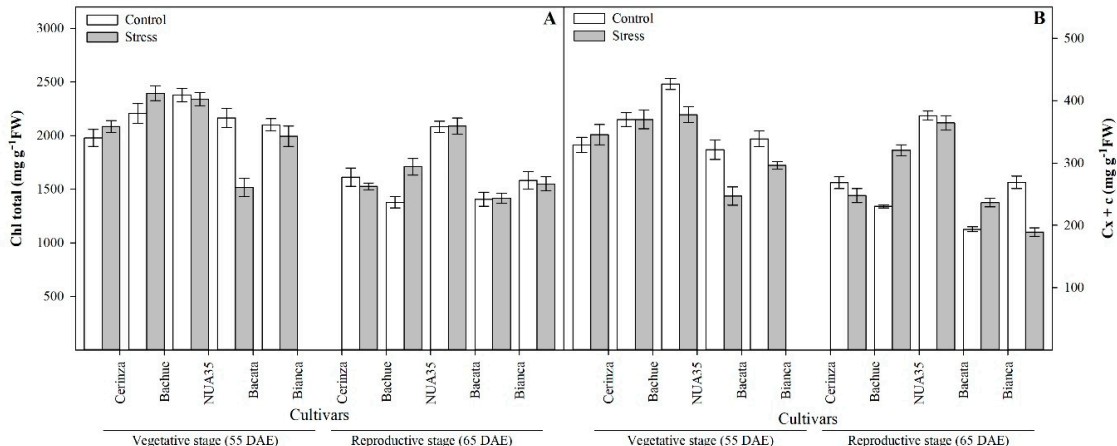

**Figure 1.** Effect of water deficit stress on two different phenological stages (vegetative (55 Days after Emergence (DAE)) and reproductive (65 DAE)) on total chlorophyll (**A**) and total carotenoids (**B**) in five bush bean cultivars. Bars represent the mean of four plants ± standard error; FW, Fresh weight.

*3.2. Fluorescence Parameters*

Table 2 presents the results for $F_v/F_m$, qP, and NPQ. Significant differences were found in the Stage × Cultivar interaction for $F_v/F_m$ and NPQ at 55 DAE. The greatest reductions in $F_v/F_m$ (0.67) and NPQ (0.71) were evidenced in cultivar NUA35 plants with a period of 15 days of stress in the vegetative stage compared to the control plants of the same cultivar ($F_v/F_m$ of 0.78 and NPQ of 1.07). Regarding qP and NPQ, significant differences were found mainly in the factor Cultivar at 55 DAE. In this regard, the lowest qP values were observed in cultivar Cerinza plants (0.56) and the highest ones in cultivar NUA35 plants (0.66). For NPQ, the lowest values were found in cultivar NUA35 plants (0.97) and the highest ones in cultivar Bacata plants (1.60). Significant differences were not obtained on qP and NPQ at 65 DAE.

**Table 2.** Effect of water deficit stress in two different phenological stages on fluorescence parameters of five bush bean cultivars (55 DAE and 65 DAE).

| Treatment | Vegetative Stage (55 DAE) | | | Treatment | Reproductive Stage (65 DAE) | | |
|---|---|---|---|---|---|---|---|
| | $F_v/F_m$ | qP | NPQ | | $F_v/F_m$ | qP | NPQ |
| Stress stage | | | | Stress stage | | | |
| Control | 0.78 a[z] | 0.61 | 1.10 | Control | 0.79 a | 0.60 | 1.07 |
| Vegetative | 0.74 b | 0.60 | 1.23 | Reproductive | 0.75 b | 0.63 | 1.21 |
| Significance | * | NS | NS[y] | Significance | NS | ** | NS | NS |
| Cultivar | | | | Cultivar | | | |
| Cerinza | 0.77 a | 0.63 | 1.12 | Cerinza | 0.77 | 0.56 | 1.27 ab |
| Bachue | 0.76 ab | 0.63 | 0.89 | Bachue | 0.75 | 0.65 | 1.10 ab |
| NUA35 | 0.74 b | 0.62 | 1.16 | NUA35 | 0.76 | 0.66 | 0.97 b |
| Bacata | 0.77 ab | 0.57 | 1.58 | Bacata | 0.75 | 0.58 | 1.60 a |
| Bianca | 0.77 ab | 0.58 | 1.29 | Bianca | 0.75 | 0.63 | 1.41 ab |
| Significance | ** | NS | NS | Significance | NS | NS | *** |
| Interaction | | | | Interaction | | | |
| Cerinza × C | 0.79 a | 0.64 | 1.22 ab | Cerinza × C | 0.79 | 0.55 | 1.47 |
| Bachue × C | 0.79 a | 0.66 | 0.60 b | Bachue × C | 0.80 | 0.65 | 0.78 |
| NUA35 × C | 0.78 a | 0.62 | 1.07 ab | NUA35 × C | 0.79 | 0.71 | 0.50 |
| Bacata × C | 0.79 a | 0.62 | 0.92 ab | Bacata × C | 0.78 | 0.54 | 1.25 |
| Bianca × C | 0.76 a | 0.51 | 1.69 ab | Bianca × C | 0.79 | 0.57 | 1.35 |
| Cerinza × V | 0.79 a | 0.58 | 1.26 ab | Cerinza × R | 0.77 | 0.57 | 0.98 |
| Bachue × V | 0.74 ab | 0.64 | 1.23 ab | Bachue × R | 0.73 | 0.65 | 1.14 |
| NUA35 × V | 0.67 b | 0.65 | 0.71 ab | NUA35 × R | 0.76 | 0.64 | 1.08 |
| Bacata × V | 0.75 a | 0.54 | 1.86 a | Bacata × R | 0.73 | 0.66 | 1.56 |
| Bianca × V | 0.77 a | 0.61 | 1.11 ab | Bianca × R | 0.77 | 0.62 | 1.27 |
| Significance | * | NS | * | Significance | NS | NS | NS |
| [x]CV (%) | 4.31 | 11.40 | 42.44 | CV (%) | 5.63 | 13.04 | 32.0 |

[z] a, b stand for the values are significantly different at $p \leq 0.05$ according to the Tukey test; [y] NS. = Not significant ($p \leq 0.05$); *, **, and *** significantly different at 0.05, 0.01 and 0.001 probability levels, respectively. [x] C.V, Coefficient of variation.

### 3.3. Rapid Light-Response Curves

RLCs showed significant differences between treatments at the end of the water deficit period in the vegetative (Figure 2) or reproductive (Figure 3) stages. A reduction of ETR values was registered with an increase of the actinic light. Cultivars Bachue and Bacata showed the greatest drops at 55 DAE under water deficit conditions in the vegetative stage, whereas cultivars NUA35 and Bacata were the most affected at 65 DAE.

Figures 4 and 5 show the variation of qP with respect to the intensity of the actinic irradiance at the end of the water deficit period in the vegetative or reproductive stage, respectively. In general, qP showed an inversely proportional behavior to the increase in actinic irradiance in both stress periods and genotypes evaluated. The water deficit stress period in the vegetative stage caused cultivars Cerinza and Bianca to show an increase in qP, while a reduction of this variable was observed in cultivar Bacata. At the end of the stress period in the reproductive stage, 'Bacata' was the only cultivar that showed a variation on qP, observing a slight increase of this variable compared to plants under control conditions.

Non-photochemical quenching showed a directly proportional tendency to the increase in actinic irradiance in the evaluated factors (genotypes and stress stage). The water deficit caused a reduction of the NPQ values in cultivars NUA35 and Bianca compared to plants under control conditions at the end of water stress in the vegetative stage (55 DAE). However, the water deficit produced an opposite tendency in cultivars Cerinza, Bachue and Bacata, in which an increase in the energy dissipation in the form of heat was observed in comparison to plants without stress (Figure 6). At 65 DAE (reproductive stage), water deficit caused two differential behaviors on NPQ in the evaluated genotypes. This variable showed a greater increase in cultivars Bachue and NUA35 under water deficit stress, whereas NPQ was lower in stressed 'Cerinza' plants (Figure 7).

Table 3 summarizes the parameters obtained from the rapid light-response curves ($\alpha$, $ETR_{max}$ and $I_k$) at 55 and 65 DAE, respectively. Significant differences were observed in the Stage $\times$ Cultivar interaction for the variables $\alpha$ and $I_k$ in the stress period in the vegetative stage (40–55 DAE). At the end of the vegetative stage, the water deficit caused a reduction of $\alpha$ values in cultivar Bacata compared to plants under control conditions (0.23 and 0.17, respectively). For $I_k$, cultivars Bacata and Bianca showed the lowest values due to the stress condition (838.19 and 769.99 $\mu mol \cdot m^{-2} \cdot s^{-1}$, respectively). Differences in the interaction of the evaluated factors were observed only on $ETR_{max}$ in the reproductive stage (50–65 DAE). The water deficit caused a reduction only on $ETR_{max}$ in cultivar NUA35, obtaining values close to 158.5 in stressed plants compared to control plants (251.22).

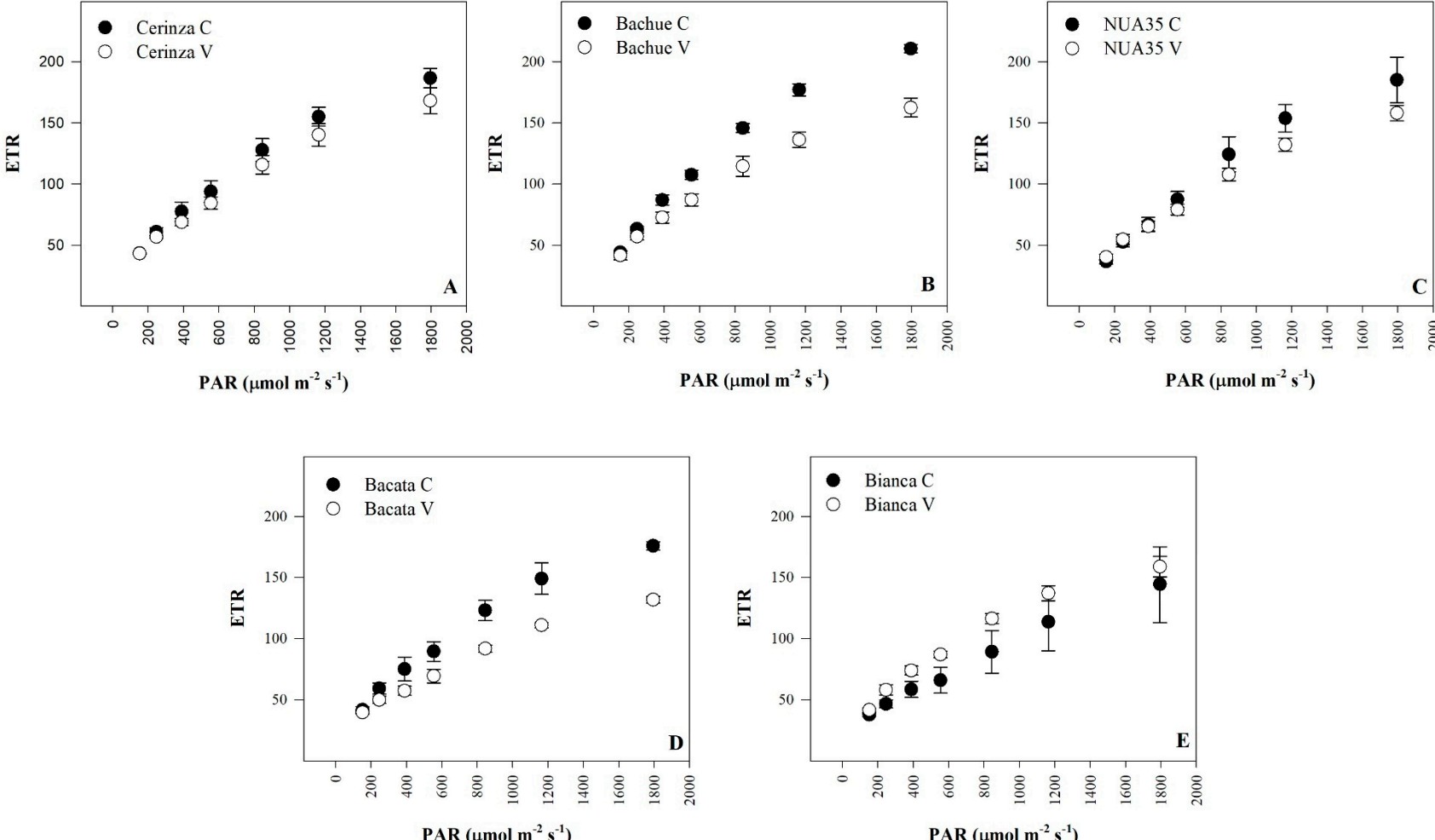

**Figure 2.** Electron transport rate (ETR) obtained from rapid light-response curves of 'Cerinza' (**A**), 'Bachue' (**B**), 'NUA35' (**C**), 'Bacata' (**D**) and 'Bianca' (**E**) under two water treatments (control (●) and water-stressed (○) plants) at the end of vegetative stage (55 DAE). Data represent the mean of four data points ± standard error; PAR, Photosynthetic Active Radiation.

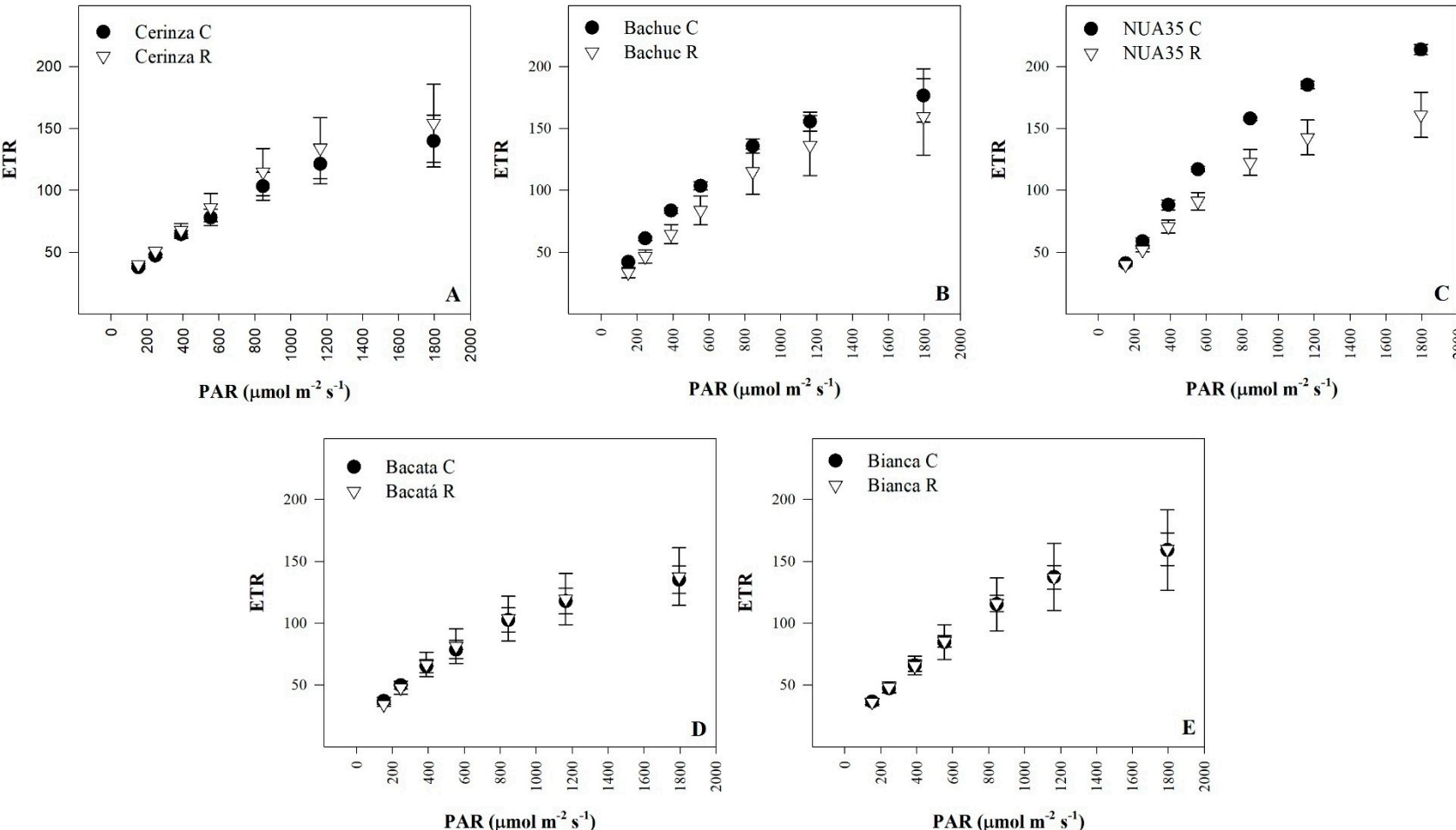

**Figure 3.** Electron transport rate (ETR) obtained from rapid light-response curves of 'Cerinza' (**A**), 'Bachue' (**B**), 'NUA35' (**C**), 'Bacata' (**D**) and 'Bianca' (**E**) under two water treatments (control (●) and water-stressed (▽) plants) at the end of reproductive stage (65 DAE). Data represent the mean of four data points ± standard error.

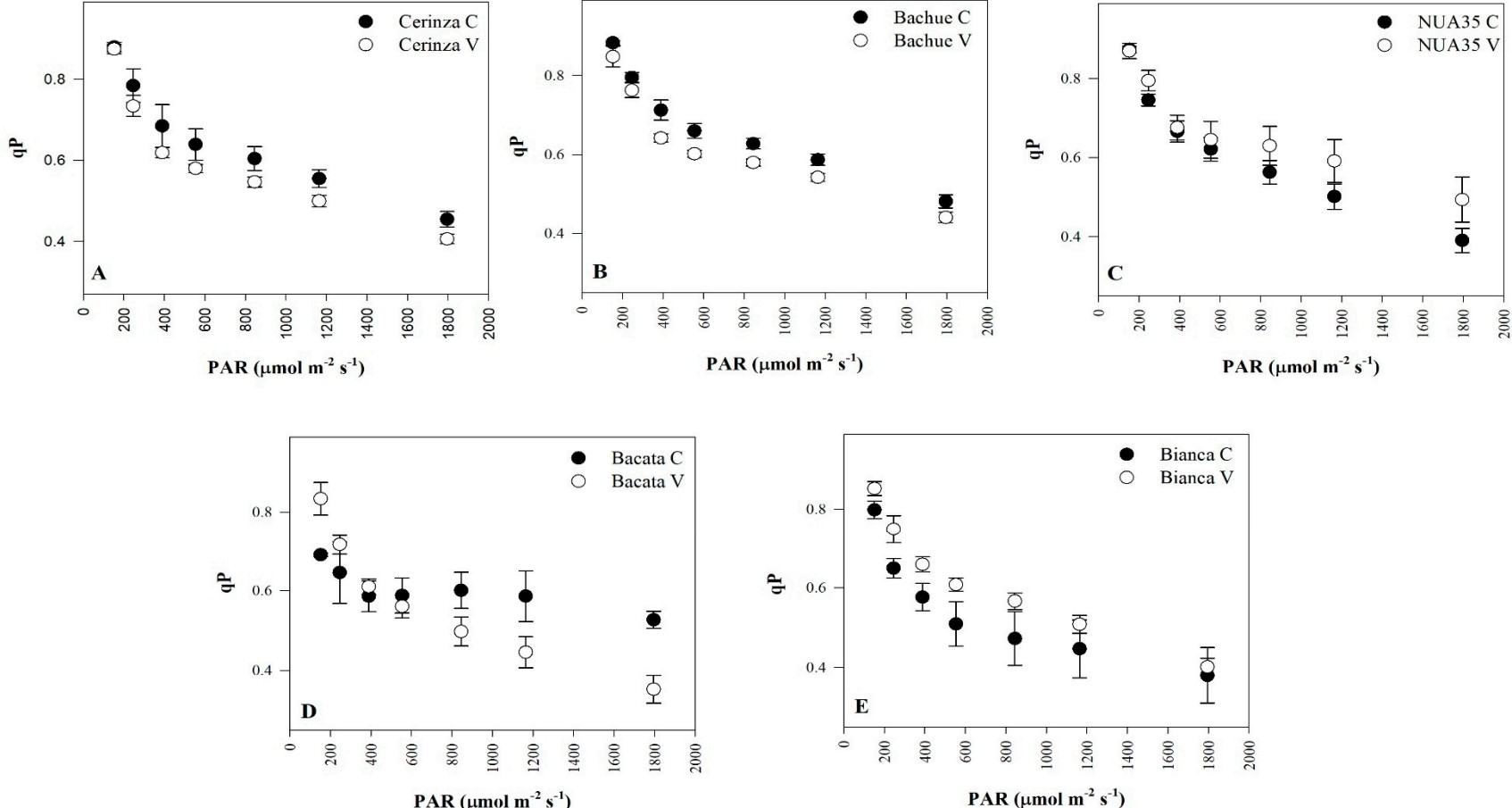

**Figure 4.** Photochemical quenching (qP) obtained from rapid light-response curves of 'Cerinza' (**A**), 'Bachue' (**B**), 'NUA35' (**C**), 'Bacata' (**D**) and 'Bianca' (**E**) under two water treatments (control (●) and water-stressed (○) plants) at the end of vegetative stage (55 DAE). Data represent the mean of four data points ± standard error.

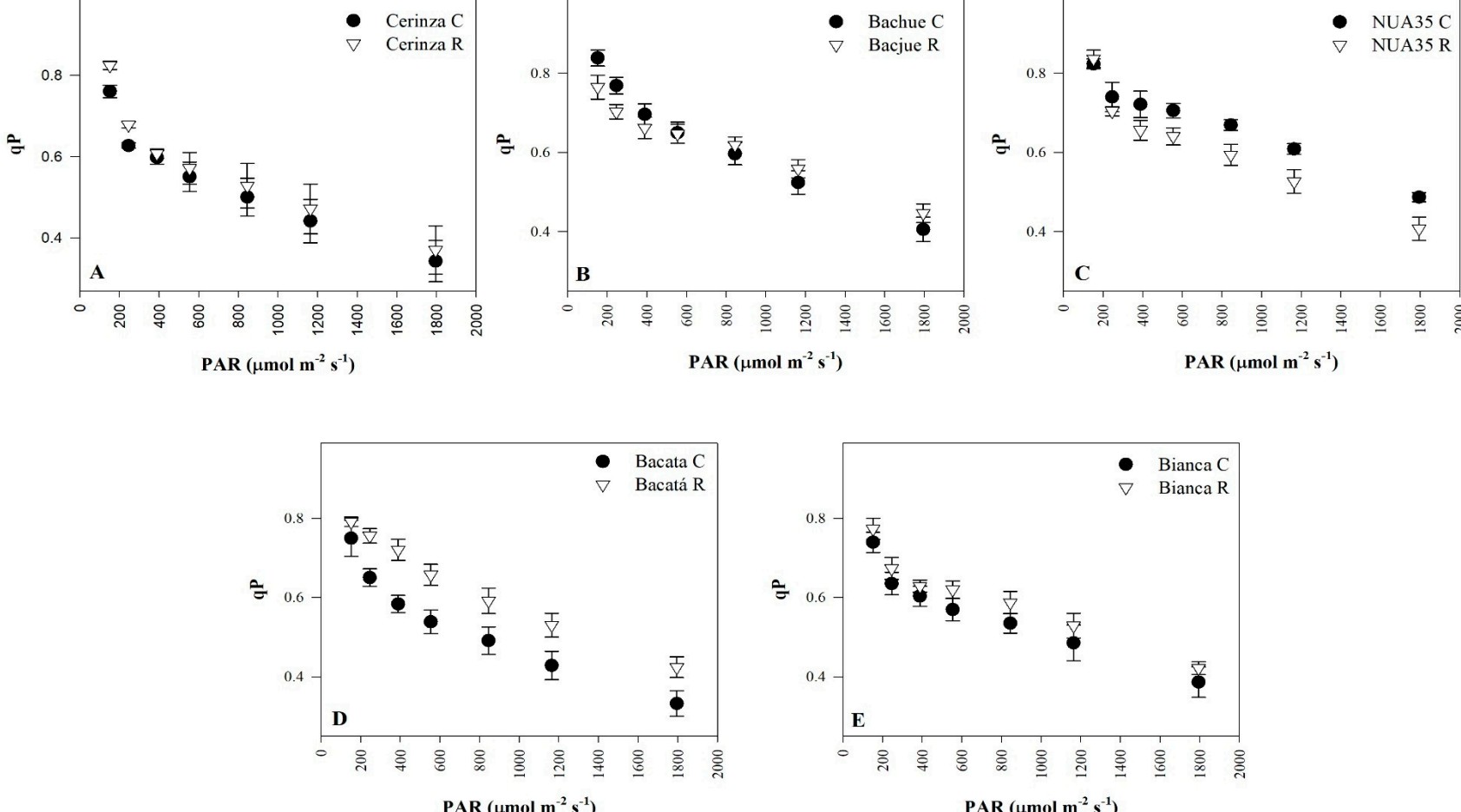

**Figure 5.** Photochemical quenching (qP) obtained from rapid light-response curves of 'Cerinza' (**A**), 'Bachue' (**B**), 'NUA35' (**C**), 'Bacata' (**D**) and 'Bianca' (**E**) under two water treatments (control (●) and water-stressed (▽) plants) at the end of reproductive stage (65 DAE). Data represent the mean of four data points ± standard error.

**Table 3.** Effect of the interaction between water stress and common bush bean (*Phaseolus vulgaris*) cultivars on the initial slope of the curve ($\alpha$), maximum electron transport rate (ETR$_{max}$), and minimum saturation irradiance ($I_K$).

| | Vegetative Stage (55 DAE) | | | | Reproductive Stage (65 DAE) | | |
|---|---|---|---|---|---|---|---|
| **Treatment** | $\alpha$ ($\mu$mol·m$^{-2}$·s$^{-1}$) | ETRmax | $I_k$ ($\mu$mol·m$^{-2}$·s$^{-1}$) | **Treatment** | $\alpha$ ($\mu$mol·m$^{-2}$·s$^{-1}$) | ETRmax | $I_k$ ($\mu$mol·m$^{-2}$·s$^{-1}$) |
| Stress stage | | | | Stress stage | | | |
| Control | 0.23 a [z] | 207.42 | 936.47 | Control | 0.24 | 190.38 a | 795.48 |
| Vegetative | 0.21 ab | 191.87 | 876.60 | Reproductive | 0.22 | 165.71 b | 811.98 |
| Significance | NS[y] | NS | NS | Significance | NS | *** | NS |
| Interaction | | | | Interaction | | | |
| Cerinza | 0.22 ab | 241.35 | 1122.83 | Cerinza | 0.22 b | 151.75 b | 768.83 ab |
| Bachue | 0.24 a | 209.75 | 896.54 | Bachue | 0.23 ab | 194.09 a | 830.91 ab |
| NUA35 | 0.22 ab | 204.39 | 916.10 | NUA35 | 0.24 a | 204.87 a | 893.33 ab |
| Bacata | 0.19 b | 169.94 | 850.35 | Bacata | 0.19 b | 150.75 b | 685.04 b |
| Bianca | 0.21 b | 203.09 | 976.53 | Bianca | 0.21 b | 188.75 a | 850.14 a |
| Significance | ** | NS | NS | Significance | * | *** | * |
| Interaction | | | | Interaction | | | |
| Cerinza × C | 0.23 abc | 232.35 | 1052.72 ab | Cerinza × C | 0.21 | 167.64 bcd | 773.82 |
| Bachue × C | 0.27 a | 221.03 | 838.06 ab | Bachue × C | 0.27 | 195.83 b | 728.18 |
| NUA35 × C | 0.24 a | 185.92 | 786.85 b | NUA35 × C | 0.28 | 251.22 a | 910.54 |
| Bacata × C | 0.23 abc | 207.85 | 884.87 ab | Bacata × C | 0.21 | 159.22 cd | 707.97 |
| Bianca × C | 0.16 c | 189.95 | 1119.85 a | Bianca × C | 0.21 | 188.56 bc | 856.91 |
| Cerinza × V | 0.21 abc | 201.21 | 953.87 ab | Cerinza × R | 0.23 | 145.86 d | 758.85 |
| Bachue × V | 0.23 ab | 196.73 | 844.59 ab | Bachue × R | 0.20 | 196.91 bc | 905.40 |
| NUA35 × V | 0.21 abc | 235.22 | 976.33 ab | NUA35 × R | 0.24 | 158.5 bcd | 774.97 |
| Bacata × V | 0.17 bc | 148.54 | 838.19 ab | Bacata × R | 0.21 | 152.89 bcd | 718.17 |
| Bianca × V | 0.23 abc | 177.65 | 769.99 b | Bianca × R | 0.21 | 188.93 bc | 902.48 |
| Significance | ** | NS | ** | Significance | NS | *** | NS |
| CV[x] (%) | 12.50 | 19.66 | 13.16 | CV[x] (%) | 14.22 | 10.21 | 13.76 |

[z] Values within a column followed by different letters are significantly different at $P \leq 0.05$ according to the Tukey test.
[y] N.S. = Not significant ($P \leq 0.05$); *, **, and *** significantly different at 0.05, 0.01 and 0.001 probability levels, respectively.
C.V[x], Coefficient of variation.

## 4. Discussion

PSII photochemistry is affected when plants are under abiotic stress conditions and the fluorescence parameters of chlorophyll $a$ are potentially useful for detecting genotypes with tolerance or susceptibility traits [19,25]. Chlorophyll $a$ fluorescence parameters (NPQ and F$_v$/F$_m$) are also considered physiological markers that provide information about the acclimation of the photosynthetic apparatus to stresses [26,27]. The present study showed that F$_v$/F$_m$ and NPQ were the parameters that showed differences between the studied factors under water deficit conditions, especially in the vegetative stage. In general, F$_v$/F$_m$ and NPQ showed a decrease in the genotypes used (Table 3). Sharma et al. [28] found that wheat genotypes susceptible to drought had lower F$_v$/F$_m$ values. Hazrati et al. [29] also reported that *Aloe vera* plants with a prolonged water deficit period (plants irrigated at 20% field capacity for two months) recorded reductions in the NPQ. The continuous decrease in these two parameters (F$_v$/F$_m$ and NPQ) indicates that the photo-protection mechanisms of energy dissipation are being affected by the stress condition, especially in cultivar Bacata, whose values presented the greatest reductions in the vegetative stage [26,30].

RLCs provide detailed information on the plant photosynthetic behavior through the study of the electron transport saturation characteristics [28]. In the present study, RLCs (ETR, qP and NPQ) better described the effects of water deficit between genotypes in the two different development stages studied compared to chlorophyll $a$ fluorescence parameters. In this sense, cultivar Bacata mainly showed the greatest variations in its RLCs, with lower ETR and qP under water deficit, whereas NPQ was generally higher in the two stress periods. Exposure to high irradiance points in RLCs activates the NPQ mechanisms to minimize photo-damage [31,32]. Serôdio et al. [33] conclude that this increase in NPQ after exposure to different actinic light intensities indicates a recovery process after a stress period.

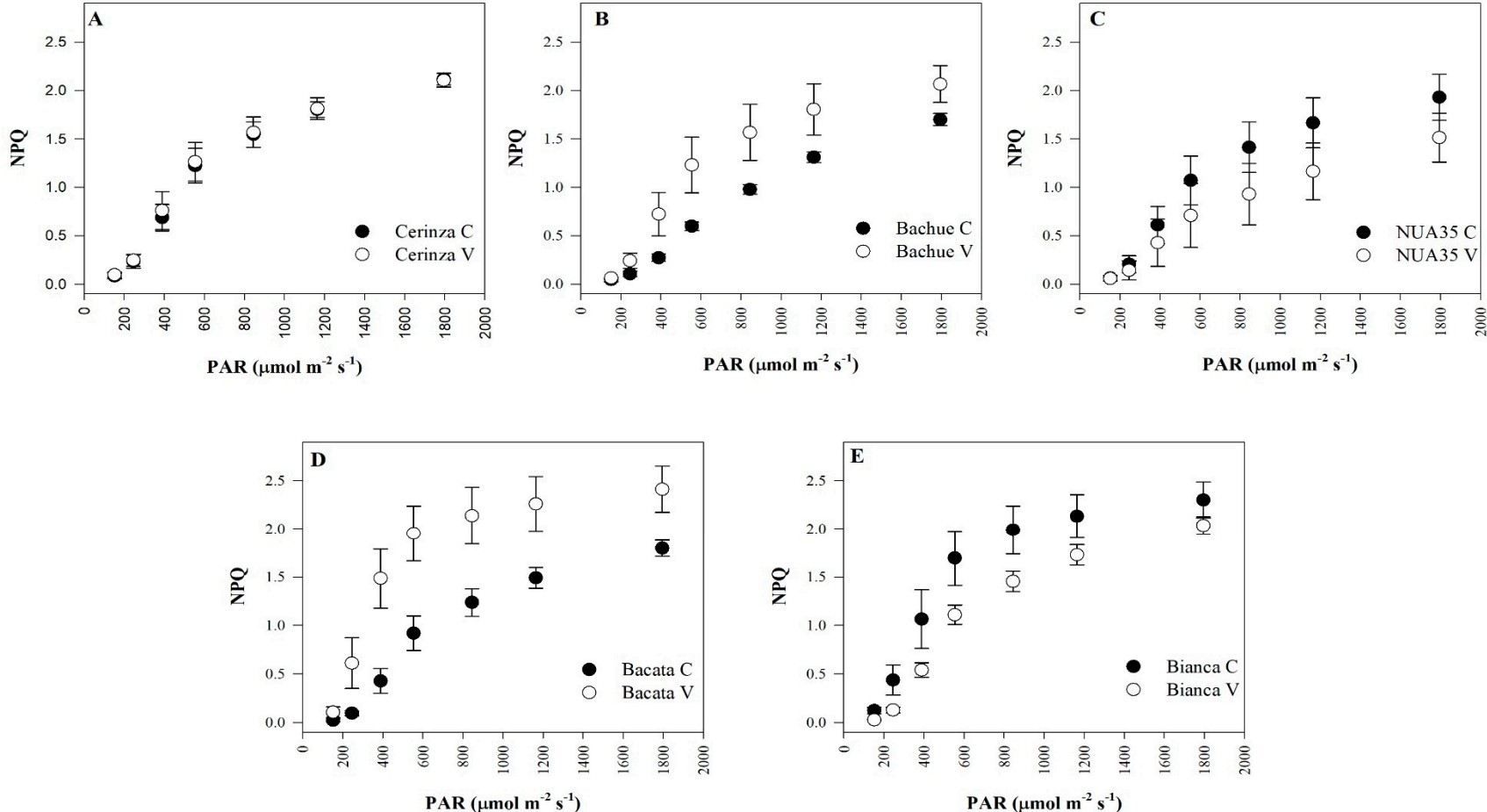

**Figure 6.** Non-photochemical quenching (NPQ) obtained from rapid light-response curves of 'Cerinza' (**A**), 'Bachue' (**B**), 'NUA35' (**C**), 'Bacata' (**D**) and 'Bianca' (**E**) under two water treatments (control (●) and water-stressed (○) plants) at the end of vegetative stage (55 DAE). Data represent the mean of four data points ± standard error.

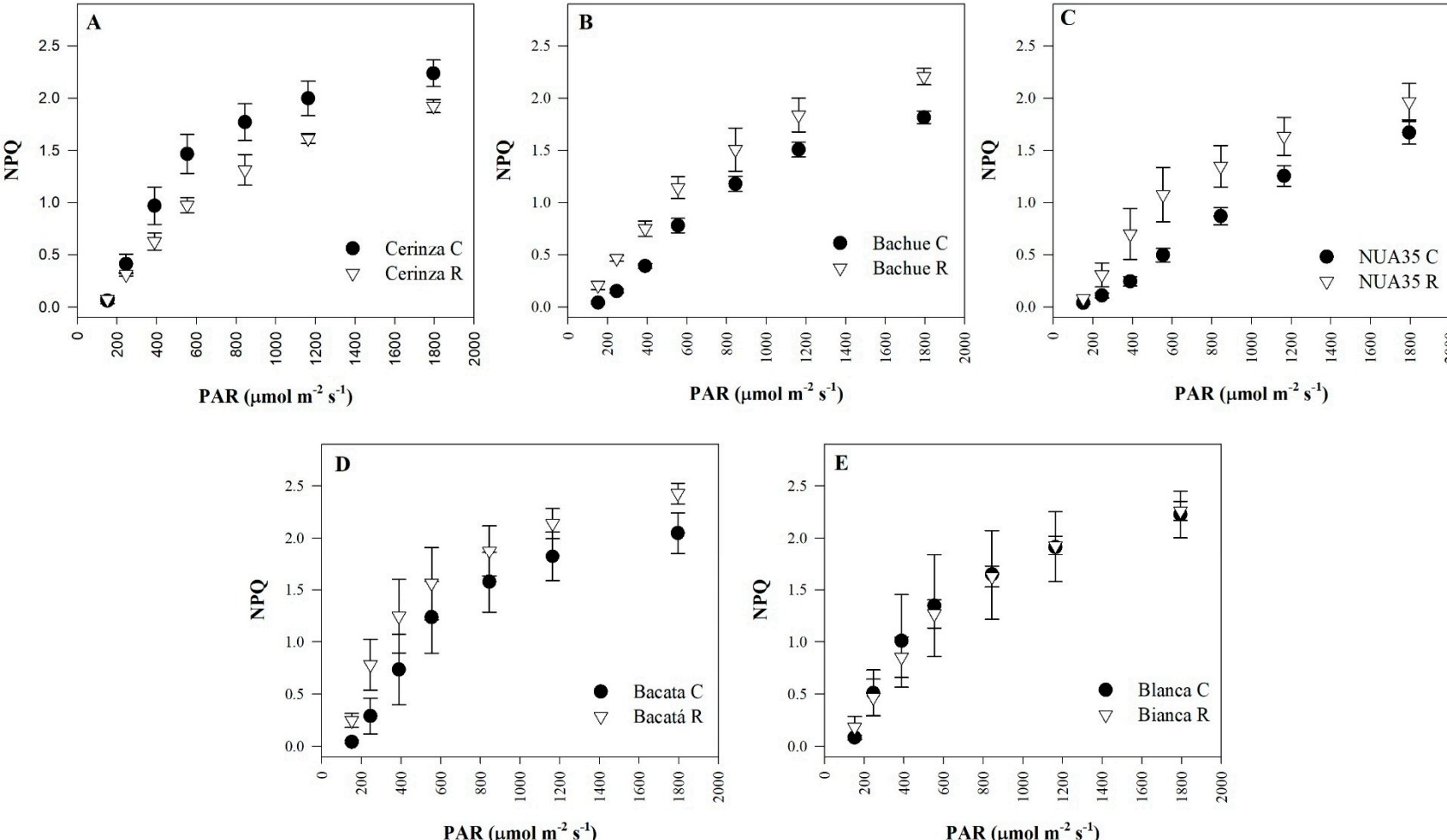

**Figure 7.** Non-photochemical quenching (NPQ) obtained from rapid light-response curves of 'Cerinza' (**A**), 'Bachue' (**B**), 'NUA35' (**C**), 'Bacata' (**D**) and 'Bianca' (**E**) under two water treatments (control (●) and water-stressed (▽) plants) at the end of reproductive stage (65 DAE). Data represent the mean of four data points ± standard error.

Photosynthetic pigments (chlorophylls and carotenoids) play a photoprotective role since they eliminate reactive oxygen species, disperse excess energy in the form of heat or suppress lipid peroxidation [34]. The obtained results showed that water deficit caused a reduction of the chlorophyll content in susceptible genotypes such as Bacata, whereas cultivar Bachue showed an increase in this variable. Reductions in the chlorophyll content have been reported in susceptible *Phaseolus vulgaris* L. genotypes to drought stress [35]. A reduction of chlorophyll under drought stress conditions is due to an overproduction of reactive oxygen species in the thylakoids [36]. In this sense, decreases in the chlorophyll content are considered a typical symptom of oxidative stress due to drought [37].

Variations in the content of carotenoids were recorded in the present work, with cultivars Bachue and Bacata showing an increase of this variable, whereas 'NUA35' and 'Bianca' showed reductions. A high carotenoid production has been reported in drought-tolerant wheat genotypes [38]. On the other hand, reductions in the carotenoid content have also been documented in *Solanum aethiopicum* and *Solanum macrocarpon* genotypes that are susceptible to water deficit [36]. Murtaza et al. [37] state that increases in pigment content help plant tolerance to drought. However, these authors also state that carotenoids present a high oxidative degradation in plants susceptible to water deficit, which is reflected in a lower foliar concentration of this pigment.

## 5. Conclusions

The parameters derived from chlorophyll $\alpha$ fluorescence ($F_v/F_m$, NPQ and $ETR_{max}$) and the development of RLCs are useful for the characterization of bush bean cultivars that are tolerant or susceptible to water deficit. In this sense, the results also support previous reports [39] in which cultivar Bacata proved to be a susceptible genotype since it showed considerable reductions in the different components of grain yield. The above suggests that PSII photochemistry is a tool that helps to understand the agronomic responses of common bush bean cultivars to drought conditions. Additionally, the photosynthetic pigment variables explain mechanisms of acclimation to water stress conditions and can be considered as a trait for the selection of genotypes in bean breeding programs.

**Author Contributions:** Conceptualization, A.D.S.-R. and H.R.-D.; Methodology, A.D.S.-R.; Software, A.D.S.-R.; Validation, A.D.S.-R., G.A.L.-M. and H.R.-D.; Formal Analysis, A.D.S.-R.; Investigation, A.D.S.-R.; Resources, G.A.L.-M.; Data Curation, A.D.S.-R.; Writing-Original Draft Preparation, A.D.S.-R.; Writing-Review & Editing, H.R.-D.; Visualization, G.A.L.-M.; Supervision, H.R.-D.; Project Administration, H.R.-D.; Funding Acquisition, H.R.-D.

**Funding:** This research received no external funding.

**Conflicts of Interest:** The authors declare no conflicts of interest.

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
