# Peer review of "Chlorophyll α Fluorescence Parameters as an Indicator to Identify Drought Susceptibility in Common Bush Bean"

_agronomy, doi:10.3390/agronomy9090526_

Round 1

Reviewer 1 Report

Line 5: why 1 differs to 2?

Line 28: delete Chlorophyll α fluorescence: it is in title; maybe add: Phaseolus vulgaris

Line 34: 55,627,164t becomes  55,627 Mt

Lines 82-83: plant -1   becomes  /plant

Line 91. add after cycle : (control)

Table 1 : on right:   %%DAE maybe 65DAE?????

Line 152: reduction : please add % value   and so on

TAble 2: check NPQ 1.1  , it becames 1.10

NUA 35XC  NPQ 1.07a : ??????, and so 1.22 ab 

abstract: the results are scarce; they can be improved

introduction: can be improved, please read: 

Rosenquist and van Kooten, 2003

Flower et al., 2007

M&M: can be improved, please read Dias & Bruggemann, 2010

Author Response

The manuscript has been corrected following all comments by peer reviewers.

Line 5: why 1 differs to 2?

Authors response: The authors affiliation was adjusted. Please, check lines 5 to 6 in the revised manuscript.

Line 28: delete Chlorophyll α fluorescence: it is in title; maybe add: Phaseolus vulgaris

Authors response: The keywords were adjusted. Please, check line 33 in the revised manuscript.

Line 34: 55,627,164t becomes  55,627 Mt.

Authors response: The value and units were corrected. Please, check line 38 in the revised manuscript.

Lines 82-83: plant -1  becomes  /plant

Authors response: plant -1   became  /plant. Please, check line 95 in the revised manuscript.

Line 91. add after cycle : (control)

Authors response: The word “control” was added. Please, check line 104 in the revised manuscript.

Table 1 : on right:   %%DAE maybe 65DAE?????

Authors response: The days were corrected in the Table 1. Please, check line 170 in the revised manuscript.

Table 2: check NPQ 1.1  , it becames 1.10

NUA 35XC  NPQ 1.07a : ??????, and so 1.22 ab 

 Authors response: Values were corrected, Please, check line 206 in the revised manuscript.

Abstract: the results are scarce; they can be improved

 Authors response: The abstract was improved and the results were included. Please, check lines 10 to 32 in the revised manuscript.

Introduction: can be improved, please read: 

Rosenquist and van Kooten, 2003

Flower et al., 2007

 Authors response: The introduction was improved by adding those references. Please, check lines 58 to 64 in the revised manuscript.

M&M: can be improved, please read Dias & Bruggemann, 2010

Authors response: We improved the description of materials and methods following the suggestion by reviewer. Please, check lines 139 to 146.

Reviewer 2 Report

Several figure legends indicate three water deficit treatments while the figures actually show control and a single treatment; the third treatment is actually water deficit imposed at different phenological stages.  This is confusing. The authors should be aware that better methods are available for determining chlorophyll and carotenoid contents.  For future research, see Sims and Gamon, Remote Sensing of Environment 81 (2002) 337-354. Is a total water limitation for 15 days realistic for the climate in which the authors live?  the authors should consider a follow up study in which a partial water limitation is imposed for that length of time.

Author Response

The manuscript has been corrected following all comments by peer reviewers. 

Several figure legends indicate three water deficit treatments while the figures actually show control and a single treatment; the third treatment is actually water deficit imposed at different phenological stages.  This is confusing.

Authors response: figure legends were adjusted. Please, check the revised manuscript.

The authors should be aware that better methods are available for determining chlorophyll and carotenoid contents.  

Authors response: The authors thank the reviewer suggestion. However, we have worked with this technique in previous studies.

Is a total water limitation for 15 days realistic for the climate in which the authors live?  the authors should consider a follow up study in which a partial water limitation is imposed for that length of time. 

Authors response: This period was defined based on previous studies carried out by the authors and explain this in the manuscript. Please, check lines 108-109.

Round 2

Reviewer 1 Report

The subject dealt in the manuscript undoubtedly falls within the general scope of the journal and as far as I know it can be considered as an interesting contribution, though not completely original, to this  field of research.

The scientific quality of the contribution is generally good, and the paper is clear in the objectives and  materials and methods.
The number and the quality of references witness a good literature